# Stem Cell Theory of Cancer: Implications of a Viral Etiology in Certain Malignancies

**DOI:** 10.3390/cancers13112738

**Published:** 2021-06-01

**Authors:** Shi-Ming Tu

**Affiliations:** Department of Genitourinary Medical Oncology, Unit 1374, The University of Texas, MD Anderson Cancer Center, 1155 Pressler Street, Houston, TX 77030-3721, USA; stu@mdanderson.org; Tel.: +1-(713)-563-7268; Fax: +1-(713)-745-1625

**Keywords:** cancer stem cell, COVID-19, HBV, HPV, SARS-CoV-2, autoimmunity

## Abstract

**Simple Summary:**

We postulate that a virus is more likely to cause cancer when it infects a progenitor stem-like cell rather than a progeny differentiated cell. We propose that the virus may turn out to be a surreptitious agent and a serendipitous model in our quest to investigate the origin of cancer. When it pertains, oncology recapitulates ontogeny, although genetic makeup is king. Cellular context may be the key to elucidating a stem cell origin of cancer.

**Abstract:**

In 1911, Peyton Rous (Nobel Prize winner in 1966) demonstrated that a virus (i.e., RSV) caused cancer in chickens. In 1976, Bishop and Varmus (Nobel Prize winners in 1989) showed that the cellular origin of retroviral oncogenes was actually normal cellular genes (i.e., proto-oncogenes). In this article, we revisit the role viruses play in the genetic origin of cancer. We review a link between viruses or cancer and autoimmunity in an alternative stem cell origin of cancer. We propose that a virus is more likely to cause cancer when it infects a progenitor stem-like cell rather than a progeny differentiated cell. We postulate that both known (e.g., HBV and HPV) and novel viruses (e.g., SARS-CoV-2) pose an imminent threat in the emergence of chronic viral diseases as well as virally induced malignancies. Knowing the origin of cancer has profound implications on our current conception and perception of cancer. It affects our conduct in cancer research and our delivery of cancer care. It would be ironic if viruses turn out to be a useful tool and an ideal means in our quest to verify a genetic versus stem cell origin of cancer. When it pertains, oncology recapitulates ontogeny; although genetic makeup is pivotal, cellular context may be paramount to elucidating a stem cell origin of cancer.


*“Gentlemen, it is the microbes who will have the last word.”*

*(Messieurs, c’est les microbes qui auront le dernier mot)*
Louis Pasteur

At present, many people are worried about a novel virus, SARS-CoV-2, and afraid of the strange disease, COVID-19. We are anxious about the economics of the pandemic. We are nervous about our welfare and healthcare. We are curious about what makes this virus so treacherous and dangerous.

It is true that viruses are part of life. They are ubiquitous and multitudinous. Most viruses are innocuous. Some are pathogenic. A few may be malignant.

Many of us may not realize that viruses may also cause cancer [1]. Viral infection is associated with cancer formation. We find viral DNA embedded (integrated or episomal) within the cancer genome [2]. However, it is evident that most viruses do not cause cancer. We also detect viral DNA in non-cancer cells [3].

Importantly, what makes a virus oncogenic may allude to the very origin of cancer.

Knowing the origin of cancer would have profound implications on our current conception and perception of cancer. It would affect our conduct in cancer research and our delivery of cancer care.

We postulate that a virus is more likely to cause cancer when it infects a progenitor stem-like cell rather than a progeny differentiated cell (Figure 1). We propose that the virus may turn out to be a useful tool and an ideal means in our quest to verify the origin of cancer. When it pertains, oncology recapitulates ontology; viral content may be king, but cellular context is key.

## 1. A Brief History

People may forget that the discovery of viral oncogenes laid the groundwork for a genetic theory of cancer. In 1911, Rous demonstrated that a virus (named Rous sarcoma virus or RSV) caused cancer in chickens [4]. By creating mutant viruses that no longer caused tumors, Duesberg and Vogt managed to attribute RSV’s cancer-causing ability to a single gene (i.e., an oncogene, known as SRC) in the virus [5]. In 1976, Bishop and Varmus showed that the cellular origin of retroviral oncogenes was actually normal cellular genes (i.e., known as proto-oncogenes) [6]. In 1989, Bishop and Varmus received the Nobel Prize in Physiology or Medicine “for their discovery of the cellular origin of retroviral oncogenes.”

Unbeknownst to most people, Dolberg and Bissell performed a perplexing experiment with provocative results [7]. When they injected RSV into 4-day-old chicken embryos, no tumors formed, even though there was v-Src activation and widespread RSV infection in the embryo. In contrast, the same virus was tumorigenic in newly hatched chicks. Furthermore, the infected embryonic cells did form tumors when grown outside the embryo. It seems that the embryonic niche harbored a mysterious force and harnessed a magical power that protected cells from cancer formation. Somehow, it managed to block tumorigenesis despite the expression of oncogenic v-Src.

## 2. Virally Induced Malignancies

Undoubtedly, Nature has already performed and is still performing some of the most astounding experiments to inform us about the origin and nature of cancer. In fact, Nature is providing us tantalizing proof through various viral infections that cancer is a stem cell disease and has a stem cell origin.

When stem cells are chronically irritated and irreparably damaged, malignancy almost invariably ensues. This happens in the lungs of a heavy smoker, in the colon of somebody with ulcerative colitis, and in the liver of a person with chronic hepatitis.

However, when only differentiated progeny cells are affected and the progenitor stem cells remain intact, the latter is poised to repair any badly wounded tissues and regenerate any collaterally damaged organs.

Therefore, the regulation of viral lytic and latent cycles in a progeny cell or a progenitor cell may be key for viral-induced tumorigenesis [8]. Viral lytic replication is more likely to occur in an active differentiated cell than a quiescent stem-like cell. An episomal viral genome is less likely to be lasting in a transient progeny cell than a permanent progenitor cell. We propose that both persistent infection-clonal expansion and hit-and-run carcinogenesis are less plausible in a progeny cell compared with a progenitor cell.

Hence, hepatitis B virus (HBV) harms liver progenitor cells and elicits hepatocellular carcinoma (HCC) [9,10]. HPV disturbs progenitor cells in the epidermal tissues and causes squamous cell carcinoma in the cervix and penis [11]. Epstein–Barr virus (EBV) disrupts progenitor cells in the head and neck and causes nasopharyngeal carcinoma (NPC) [12].

## 3. Stem Cell Origin of Cancer

When viruses infiltrate differentiated progeny cells, our immune cells recognize the virally infected cells and contain the viral infection. Furthermore, the dispensable and disposable differentiated cells have a limited life span. We eliminate the damaged and dead infected cells along with the viruses within them.

In contrast, when viruses sabotage progenitor cells, our immune system may fail to recognize these cells as infected. Indeed, virally infected cells derived from progenitor cells with stem-ness properties (e.g., those involving the PD-1/PD-L1 axis [13,14]) have more means to evade and elude the immune system. Progenitor cells, unlike progeny cells in multicellular organisms, exist for an extended time, if not perpetually. They are immune privileged for a good reason—we cannot afford to destroy our own seeds or deplete our very source of life.

Although a virus can attach, internalize, integrate, proliferate, and propagate in both progenitor and progeny cells for its own purposes and benefits, the type of cell it infects does make all the difference in the manifestation, continuation, and ramification of infection. In other words, when it concerns the acute versus chronic consequences of a viral infection, including its malignant potential, cellular context is paramount [15,16].

When we consider cellular context, we should not ignore or neglect the niche, which involves the immune system in an ongoing viral infection, and which no longer protects an individual from carcinogenesis when that person is no longer embryonic.

## 4. HBV and Hepatocellular Carcinoma

In many respects, HBV provides us with one of the most compelling pieces of evidence (if not proof) of a stem cell origin of cancers.

HBV causes inflammation and cirrhosis in the liver, which contribute to the development of HCC. Overall, a majority of HCC cases are attributable to persistent HBV infection.

HBV spreads through the parenteral route (e.g., blood and body fluids). It infects about one out of three people in the world. Most people with the chronic disease have no symptoms. However, about 25% of those afflicted with chronic HBV infection eventually develop cirrhosis or HCC. 

Importantly, of those infected by HBV around the time of birth, 90% develop chronic hepatitis B, while of those infected after age five, less than 10% do [17].

Perhaps most liver cells are progenitor cells in the fetus or a newborn. The embryonic cells and niche may be tolerant of HBV and do not become malignant as long as they remain embryonic. The immune system accepts an embryonic progenitor cell even it is infectious or malignant.

In contrast, the hepatitis A virus (HAV) spreads through the oral-fecal route. After a single infection, a person is immune for life. Therefore, HAV is not associated with chronic infection or HCC.

Perhaps those cells lining the portal system in the hepatic sinusoids and exposed to viruses arriving from the digestive tract versus the rest of the body are more progeny than progenitor [18]. Perhaps HAV has an affinity for progeny cells and HBV for progenitor cells by virtue of their respective locations and cell surface adhesion molecules and receptors.

Perhaps HAV could serve as a convenient negative control for the study of chronic hepatitis and HCC in this incredible experiment of Nature.

## 5. Experiments of Nature and by Man

Importantly, cirrhosis caused by alcohol is also prone to form HCC. Approximately 10–20% of heavy drinkers develop cirrhosis [19]. Furthermore, the annual incidence of HCC in patients with alcohol-induced liver cirrhosis is about 1.9–2.6% [20,21].

Perhaps alcoholic hepatitis could be an incidental positive control for the study of chronic hepatitis and HCC in this impeccable experiment of Nature.

Evidently, the virus is not the only culprit. The chronic inflammation it causes (and the eventual cirrhosis it manifests) are just as culpable. The common denominator for HCC could very well be whether the virus or the ensuing inflammation affects a progenitor cell versus a progeny cell.

Currently, we do not know whether HAV is not carcinogenic (compared with HBV) because (1) its DNA is not integrated into the host genome; (2) it does not incite chronic infection and persistent inflammation; and (3) it does not injure hepatic progenitor versus progeny cells.

Surprisingly, we humans have already performed some seminal experiments (as Nature has always done), even though we may not yet have formulated a pertinent hypothesis about a stem-cell versus genetic origin of cancer in virally induced malignancies.

For example, Holczbauer and colleagues induced the formation of hepatic cancer stem cells (CSCs) from different stages of hepatic cellular maturation, ranging from primary hepatic progenitor cells and lineage-committed hepatoblasts to differentiated hepatocytes [9]. Interestingly, CSCs originated from different cell types showed differential gene expression profiles and formed specific HCC subtypes in accordance with their cellular origins.

In addition, Anfuso et al. demonstrated that early liver inflammation caused by the accumulation of hepatitis B surface antigen (HBsAg) activated progenitor cells in a particular stem cell population [10]. Although HBsAg was an important oncogenic factor, the progressive increase of CSC markers suggested that cellular context was pivotal in the pathogenesis of HCC.

## 6. HPV and Cervical Cancer

Similarly, we have established that HPV is the etiologic agent for essentially all cervical cancers. However, it is important to point out that the majority of cervical HPV infections do not lead to cervical cancer. In fact, only 1% of HPV cases progress to high-grade dysplasia and cancer.

Therefore, almost all cervical HPV infections are transient non-neoplastic viral infections that disappear within a year rather than chronic illnesses that eventually result in cancer formation.

Again, what is responsible for this discrepancy in the prevalence of infection and the incidence of malignancy? Persistent HPV infection is a prerequisite for the development of cervical neoplasia. Importantly, persistent viral infection may only occur in specific cell types. 

Hence, when HPV infects a progeny differentiated cervical cell, the affected cell dies and disappears along with the viruses (similar to HAV). However, when HPV infects a progenitor cervical cell with stem-like properties, the infection may perpetuate and malignancy eventuate (similar to HBV) [11].

## 7. EBV and Nasopharyngeal Carcinoma

EBV is associated with several lymphoproliferative malignancies, including Burkitt lymphoma and Hodgkin’s disease, as well as with non-lymphoproliferative malignancies such as NPC and gastric cancers.

Although EBV infections are widespread and frequent in the human population, cancer formation is relatively rare and sporadic. Again, this suggests that infection by itself does not necessarily portend cancer. However, we suspect that infection of certain special cell types predisposes to cancer formation. Specifically, persistent infection and especially infection in persistent cells (rather than perishable cells)—namely, progenitor cells (rather than progeny cells)—are more prone to cancer formation.

Again, it is evident that NPC originates from clonal EBV-infected basal stem cells [12]. Hence, EBV-infected cells in some precancerous lesions only occur in or immediately adjacent to the basal layer of the nasopharyngeal epithelia. There is a significant increase in the copy number of EBV DNA and the expression of EBV latent genes in isolated NPC CSCs, suggesting that the modulation or maintenance of stem-ness properties, if not stem cells themselves, may be critical in the initiation and progression of EBV-mediated NPC.

## 8. Viral Oncogenesis

The possibility of viral infection or viral oncogenes (e.g., HPV E6/E7, EBV LMP1/2, HBx) inducing cancer stem cell properties is of interest [8]. It suggests that cellular context (affected progenitor cell versus progeny cell) is crucial because stem-ness is a multicellular phenomenon that connects the extracellular microenvironment (e.g., inflammatory response) with intracellular aberration (genomic instability).

Hence, HPV E6/E7 oncoproteins degrade and inactivate the tumor suppressors TP53 and RB1, respectively, and interfere with stem-ness pathways, including Notch and TGF-beta signaling. EBV LMP1/2 upregulate oncogenic cellular proteins and microRNAs, downregulate tumor suppressor expression, and activate signaling pathways, such as the NF-kB pathway, that promote carcinogenesis. In addition, HPV E6/E7 and HBx induce mitotic abnormalities and cause chromosome missegregation. When a viral entity disrupts a defining stem cell property (namely asymmetric cell division), a common cancer hallmark ensues (e.g., genomic instability in the form of aneuploidy).

Intriguingly, comparative genomic analyses of HPV-positive and negative head and neck cancers have demonstrated that virus-induced tumors tend to carry fewer cellular mutations than the virus-uninvolved cases [8]. Furthermore, the mutations in non-viral tumors activate or inactivate the very same pathways that the viral oncogenes disrupt, indicating that the etiology of these malignancies is ultimately analogous.

Because progenitor cells already possess abundant stem-like capabilities (e.g., migration, multipotency, dormancy) compared with their progeny counterparts, we assume that they may need fewer genetic mutations to become malignant (e.g., metastasis, heterogeneity, drug resistance). Because HPV-positive and negative head and neck cancers are clinically different diseases with clinically disparate outcomes (e.g., response to treatment and overall survival), we propose that the etiology of this difference may be in their cellular origin, if not in their genetic makeup.

## 9. Viruses and Autoimmunity 

Therefore, it is plausible that cellular context (e.g., the involvement of a progenitor stem-like cell versus a progeny differentiated cell) determines whether a viral infection has the potential to cause a malignancy. Intriguingly, it may also dictate whether a viral infection has the potential to cause autoimmunity.

Autoimmune disease occurs when the immune system attacks “self” rather than “non-self.” In the USA, about 1 in 30 or >8.5 million individuals suffer from an autoimmune condition. Understandably, autoimmunity is a common problem that generates interest and instigates inquiry.

An association between viral infections and autoimmune disease is more than circumstantial.

Yoon et al. isolated Coxsackievirus from the pancreas of a patient who died of acute diabetes [22]. Forrest et al. associated congenital rubella virus infection with the development of diabetes [23]. It is evident that various infectious agents cause and exacerbate a majority of relapsing–remitting multiple scleroses [24]. Perhaps viruses encoding cross-reacting epitopes (i.e., molecular mimicry) could surreptitiously prime a certain person for a particular autoimmune disease, and subsequently the same or similar viral infections could activate and exacerbate autoimmunity [25].

When it concerns an underlying mechanism for the autoimmune response, does it matter whether a specific molecular mimicry that activates certain autoreactive T or B cells in a susceptible individual involves unique self-peptides in a progenitor stem-like cell versus a progeny differentiated cell? Does it matter whether the invading viruses mostly infect progenitor stem-like cells and/or a progeny differentiated cells? We speculate that in a progeny differentiated cell the invading viruses only cause acute transient infections, but in a progenitor stem-like cell, they may also induce chronic persistent autoimmunity (Figure 1).

## 10. Cancer and Autoimmunity

Similarly, an association between neoplasms and autoimmune diseases is more than conjectural [26,27].

Effective immunotherapy may elicit an immune response to tumor antigens as well as to related stem cell antigens, the latter of which leads to an autoimmune reaction. Presumably, injury to differentiated cells is reparable or reversible, but injury to stem cells would cause lasting if not permanent damage and sequelae. 

Autoimmunity often arises after the successful immunotherapy of some cancers. For example, a positive thyroid autoantibody titer was highly correlated with increased survival in patients with renal cell carcinoma who received interleukin-2 (IL-2) and interferon-alfa 2 therapy [28]. Most patients who experienced substantial regression of their metastatic melanoma after treatment with tumor-infiltrating lymphocytes and IL-2 also developed anti-melanocyte autoimmunity [29]. Phan et al. [30] observed objective clinical responses in three patients with melanoma who received a CTLA-4-blocking monoclonal antibody in combination with a peptide vaccine. Among the treated patients, 43% experienced some type of grade 3 or 4 immune-mediated toxic effects, including dermatitis, enterocolitis, hypophysitis, uveitis, and hepatitis. Similarly, Beck et al. [31] reported a 14% rate of response but a 21% rate of enterocolitis in patients with renal cell carcinoma or melanoma who received a human anti-CTLA-4 monoclonal antibody over a 3-year period with or without a peptide vaccine. The patients who developed enterocolitis also happened to be those who had an objective clinical response. 

Therefore, one could account for an association between cancer and autoimmunity according to a stem cell origin of cancers and explain the phenomenon of cancer immunity and autoimmunity by potential cross-reactions between antigens shared by a malignant cell and its related progenitor stem-like cell rather than progeny differentiated cell.

## 11. Connecting the Dots

Perhaps it is not a coincidence that certain viral infections predispose one to carcinogenesis and/or autoimmune diseases. Perhaps it is not serendipity that cancer immunity and autoimmunity direct us towards a stem cell origin of cancers.

After all, it is imperative for the immune cells to avoid attacking and destroying our own healthy cells. They leave alone those cells tagged with immune checkpoints (like identity cards), which brand the cells as self rather than non-self, thereby ensuring the protection and preservation of our very own progenitor stem cells and progeny differentiated cells in the body.

Importantly, immune checkpoints enable immune cells to recognize and reject cells damaged by viruses, which attempt to alter the identity and function of affected cells to serve the viruses’ purposes instead.

When this immune safeguard malfunctions or becomes dysfunctional, the immune system attacks normal cells and tissues instead, causing havoc and misery in the form of autoimmune diseases.

Unfortunately, if normal stem cells are innately concealed or shielded from the immune system, so will be cancer stem cells that mimic them (being derived or converted from them) [15,16]. Perhaps we can outwit and outduel cancer immunity by unmasking and uncovering cancer cells for immune surveillance. The challenge is that if cancer has a stem cell origin and nature, immunotherapy could be a Sisyphean battle and a Pyrrhic victory, because a majority of refractory cancers are likely to be more self than non-self.

Perhaps this is a reason why one of the more effective immune checkpoint inhibitors (against PD-1/L1) is also one of the better immunotherapy options (compared with those against IDO1 and CD47 [32,33]). After all, PD-L1 is an immune checkpoint for certain normal stem cells [34,35] and their corresponding cancer stem cells [36]. It modulates epithelial–mesenchymal transition (EMT) and CSC-like phenotype, in which there is co-amplification of PD-L1 along with other stem-ness factors, such as MYC, SOX2, n-cadherin, and SNAI1 [37]. In addition, EMT transcriptionally induces N-glycosyltransferase STT3 through beta-catenin, and subsequent STT3-dependent PD-L1 stabilization and upregulation in CSCs more than non-CSCs [38].

On the other side of the coin, one would predict and should even expect that any successful immunotherapy (or non-immunotherapy, such as anti-Nectin-4 (poliovirus receptor-like 4 or PVRL4), anti-Trop-2 (trophoblast cell surface antigen 2), et cetera) that targets CSCs versus non-CSCs may also engender and entail nagging autoimmune complications.

## 12. Et Tu, SARS-CoV-2?

We are in the midst of a SARS-CoV-2 pandemic. Our immediate tasks are to mitigate the contagion and attenuate the calamity this virus has provoked and precipitated. We are still trying to learn its biological itineraries and clinical trajectories.

For us oncologists, the specter of virally induced malignancies still lurks in our memory of RSV and lurks in the shadows of HBV and HPV.

If SARS-CoV-2 does affect progenitor cells in various tissues, is acute COVID-19 the only problem we need to worry about, or do we also need to contend with certain latent issues, including chronic viral infections and virally induced malignancies, in the foreseeable future?

Indeed, the problem is already here with autoimmune complications. We are observing a disturbing trend of myriad autoimmune presentations caused by SARS-CoV-2, in the forms of type I diabetes mellitus and Kawasaki disease, Guillain-Barre syndrome, anti-phospholipid syndrome, et cetera [39]. We surmise that it is only a matter of time before one links the worst cases of COVID-19 to SARS-CoV-2 infection in progenitor stem cells and/or progeny differentiated cells somewhere in the body.

Understandably, a link of SARS-CoV-2 with cancer is weak. After all, SARS-CoV-2 is a new virus, and carcinogenesis takes time to become manifest.

However, Tutuncuoglu and colleagues have suggested that several SARS-CoV-2 proteins might behave like oncoproteins [40]. For example, NSD2 is involved in NF-kB activation and plays a pivotal role in the release of proinflammatory cytokines. It also enhances RAS-driven oncogenic responses. Protein E and NSD2 interact with the BET family proteins BRD4 and BRD2, both of which mediate hyperactivation of oncogenes such as MYC. Importantly, BRD4 and BRD2 regulate essential stem-like processes such as EMT and cellular plasticity [41].

SARS-CoV-2 also removes host-specific miRNAs that modulate specific gene expression to suppress immunity or prevent the activation of unfolded-protein-response-related apoptosis [42]. For example, SARS-CoV-2 suppresses 28 unique miRNAs, many of which are putative tumor suppressors—their reduced levels correlate with poor cancer prognosis. Specifically, one of the affected miRNAs is miR-34a, which is a master regulator of asymmetric division [43]. Low miR-34 level is a marker of a late-stage CSC phenotype [44].

It would be ironic if COVID-19 ultimately verifies if not proves a stem cell versus genetic origin of cancer in both virally and non-virally induced malignancies.

## 13. Conclusions

This article is a tribute to our predecessors and pioneers, who advanced our knowledge about a genetic origin of cancer in virally induced malignancies [15,16].

In the same spirit of HBV, HPV, and EBV, and in our current climate of SARS-CoV-2, we revisited a viral origin of cancer, reassessed a genetic origin of cancer, and ruminated about a stem cell origin of cancer.

In all likelihood, we are not dealing with a viral origin or genetic version of cancer, because the virus itself and the infection it causes do not cause cancer when the cells affected are short-lived progeny cells. However, we postulate and propose that when long-lived progenitor cells are involved in the viral infection, not only is the infection more likely to be chronic, it is also likely to be carcinogenic (Figure 1).

Certainly, when we consider COVID-19, the virus may have the last word. Currently, because we endorse a genetic theory of cancer, the cancer genes speak loud and clear. Eventually, when we embrace a stem cell theory of cancer, those same genes will speak in a different tongue with a different meaning. In this case, the last word may not only alter a whole world of virally induced malignancy, but also usher in a new era of stem-ness-related malignancy, without the guise of an infectious or genetic etiology that conceals and confuses its proper cellular context.

## Figures and Tables

**Figure 1 cancers-13-02738-f001:**
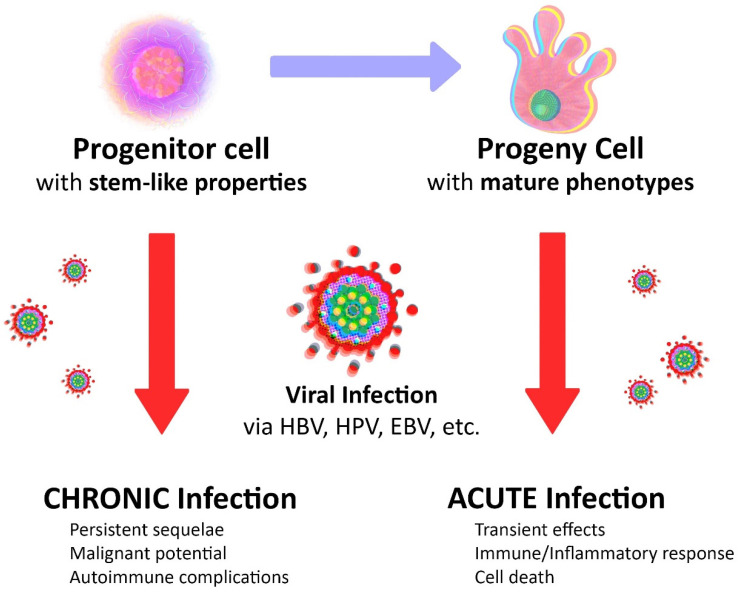
Viruses and cancer. A virus is more likely to cause malignant tumors, chronic infections, and autoimmune complications when it infects a progenitor cell with stem-like properties rather than a progeny cell with mature phenotypes (illustrated by Benjamin Tu).

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
