# Peer review of "Stem Cell Theory of Cancer: Implications of a Viral Etiology in Certain Malignancies"

_cancers, 2021, doi:10.3390/cancers13112738_

Round 1

Reviewer 1 Report

This manuscript presents the author's perspective on the cancer etiology, which is that cancer originates from stem cells and viruses can only cause cancer when infecting the progenitor cells, not their progeny. In the view of the author, viruses support the stem-cell etiology of cancer as the only viruses that lead to cancer are those, that infect the stem-like progenitor cells, eg. HBV or EBV, in contrast to eg. HAV. This is a thesis that would definitely benefit from the more in-depth argumentation and documentation, while the author provides a rather modest and superficial reasoning, not very convincing. 

  • For example, the statement: "virally infected cells derived from progenitor cells with stemness properties have more means to evade and elude the immune system" [lines 90-91] could be further explored and referred to original works to make it stronger.

  • There is nothing mentioned on the genetic programs affected in progenitor cells by specific viruses that lead to tumorigenesis (which, how and why more effective than in normal cells)

  • In a number of sections, where the author discusses key arguments for his perspective, there are hardly any references (eg. 2. Virally induced malignancies, Stem-cell origin of cancer,4,6,7), which is surprising, especially in the context of what is said in the conclusions, that: “This article is a tribute to our predecessors and pioneers, who advanced our knowledge about a genetic origin of cancer in virally induced malignancies.” References to original works, the conclusions of which are presented by the author, are essential in a work like this.

  • Generally, it’s challenging to digest the writing style used in this MS. Grammatical slips in combination with a colorful and literary language used by the author, often make the message unclear. Just a few examples:

When it pertains oncology recapitulates ontogeny, although genetic makeup is king cellular context may be the key to elucidate a stem-cell origin of cancer [lines 11-12]

Knowing the origin of cancer has profound implications on our current conception  and perception of cancer. It affects our conduct in cancer research and our delivery of  cancer care [lines 43-44]

The common denominator for HCC could very well be whether the virus or the ensuing inflammation affects a progenitor cell versus a progeny cell. [lines 135-136]

Reviewer 2 Report

This is an interesting article for discussing the potential link between viral infection and stem-cell origin of cancer. It would be better to have a figure to illustrate the hypothesis.

Minor points:

  1. Page 1, lane 39-40: “We find viral DNA embedded within the cancer genome” Does it mean integration of viral DNA into host genome? In EBV- and KSHV-associated malignancies as some HPV-associated cervical cancers, the viral genome is episomal and not integrated into the cancer genome.
  2. The regulation of viral lytic and latent cycles in the differentiated progeny and progenitor cells may be a key for viral-induced tumorigenesis. It will be more importance in the cancer with episomal viral genomes. The link between cell differentiation and viral lytic replications can be discussed. Persistent latent infection is needed for clonal expansion of viral infected cells and further cancer progression.
  3. In the manuscript, relevant references should be cited to support the statements, for example, page 1 (lane 80-83), or clearly state that it is perspective. In page 4, lane 181, the reference for “NPC originates from clonal EBV-infected ‘basal stem cells’.” should be provided. It is not sure some sentences are based on the published work or just a hypothesis.
  4. The possibility of viral infection or viral oncogenes (e.g. EBV-encoded LMP1/LMP2) induces cancer stem cell properties can be discussed.
  5. The link of SARS-CoV-2 with cancer is weak. The hypothesis is mainly based on its effect on autoimmune presentations. Since most of the cancer virus contains viral oncogenes (e.g. E6/E7 in HPV, LMP1/LMP2 in EBV, HBX in HBV..), any potential viral oncogene in SAR-CoV-2 for driving cancer formation?
  6. In page 6, lane 265, “STT3” is not Stat-3.

Round 2

Reviewer 1 Report

The authors have addressed the concerns.